# Determination of the Residue Behavior and Risk Assessment of Chlorfluazuron in Chinese Cabbage, Kale, Lettuce and Cauliflower by UPLC-MS/MS

**DOI:** 10.3390/ijerph16101758

**Published:** 2019-05-17

**Authors:** Mingna Sun, Zhou Tong, Xu Dong, Yue Chu, Mei Wang, Tongchun Gao, Jinsheng Duan

**Affiliations:** 1Institute of Plant Protection and Agro-Product Safety, Anhui Academy of Agricultural Sciences, Hefei 230031, China; sunmingna@126.com (M.S.); tongzhou0520@163.com (Z.T.); dongxu929@163.com (X.D.); chuychu@163.com (Y.C.); wangmeinc@sina.com (M.W.); 2Key Laboratory of Agro-Product Safety Risk Evaluation (Hefei), Ministry of Agriculture, Hefei 230031, China

**Keywords:** residue behavior, UPLC-MS/MS, chlorfluazuron, vegetables

## Abstract

Chlorfluazuron is used as a highly effective insect growth regulator to control a variety of crop pests. However, residues of this pesticide have been shown to be harmful to human health. To evaluate the residual dissipation pattern and risk for dietary intake of chlorfluazuron in various vegetables, a solid phase extraction-ultra performance liquid chromatography-tandem mass spectrometry method was established to analyze chlorfluazuron residues in Chinese cabbage, Chinese kale, Chinese lettuce, and cauliflower. The sample was extracted with acetonitrile and purified using an SPE amino column. The average recovery of the target sample in the analyzed four vegetables was between 75.0% and 104.1%, and the relative standard deviation was between 2.5% and 9.6%. The precision and accuracy of the analysis met the requirements of residue analysis standards. Dissipation kinetic testing of chlorfluazuron in different vegetables showed a half-life of 2.4–12.6 days, with a rapid dissipation rate. The estimated daily intake of the chlorfluazuron was 0.753–1.661 μg/(kg bw·d), and the risk quotient was 0.15–0.35. It showed that chlorfluazuron had a low risk of chronic dietary intake from vegetables in different populations in China. The results of this study has described the degradation rate of chlorfluazuron in four vegetables, evaluated the risk of dietary exposure to Chinese residents. Therefore, it provides supporting data and empirical basis for guiding the reasonable use of chlorfluazuron in vegetable production and in evaluating its dietary intake risk in vegetables.

## 1. Introduction

Chlorfluazuron is a synthetic inhibitor of insect chitin that disrupts normal exuviation by inducing malformations, thereby preventing and controlling *lepidopteran* pests [1,2]. In China, chlorfluazuron has been registered and used in crops, including vegetables, citrus, and cottons. However, the irrational application of pesticides has induced pests to develop resistance [3] and increased pollution of chlorfluazuron residues in agricultural products, including vegetables, which poses a potential threat to human health. Therefore, it is necessary to study the residual behavior of chlorfluazuron in vegetables and to perform risk assessment of dietary intake.

Numerous reports have described different analytical methods for chlorfluazuron residues in environmental samples. Nano-sized TiO_2_ has been used in the pretreatment of pesticides, including chlorfluazuron in water samples [4]. Aluminosilicate modified by β-cyclodextrin has been employed to remove the interference of matrix in the detection of chlorfluazuron in honey samples [5]. In addition, residual analysis methods of chlorfluazuron in apple [6], strawberry [7], pear [8], and tea [9] have also been established. Chen et al. presented results on two pyrethroid pesticides and three benzoylurea pesticides in tea [10]. Ganguly et al. presented results on the persistence of chlorfluazuron in cabbage and its risk assessment [11]. However, the residual behavior and risk assessment of chlorfluazuron from vegetables in China have not been reported to date. Here, we describe an UPLC-MS/MS (ultra-high-performance liquid chromatography-tandem mass spectrometry) residue analysis method for chlorfluazuron in various vegetables (Chinese cabbage, Chinese kale, Chinese lettuce, and cauliflower) was established. Furthermore, we built a SPE (solid phase extraction)-UPLC-MS/MS method for the determination of chlorfluazuron, and investigated the dissipation kinetics and performed risk assessment of dietary intake by implementing the standardized field residue test.

## 2. Materials and Methods

### 2.1. Reagents and Standards

Chlorfluazuron standard (purity 99.9%) was purchased from Dr. Ehrenstorfer Gmbh (Augsburg, Germany). C18, Florisil (magnesium silicate), NH_2_, Carbon, Alumina-N solid phase extraction cartridge (300 mg/3 mL) were purchased from Agilent Technologies Co., Ltd. (Tianjin, China). Sodium chloride and acetonitrile (analytical pure) were provided by Sinopharm Chemical Reagent Co., Ltd. (Shanghai, China). Formic acid (chromatography pure) was purchased from TEDIA (Shanghai, China).

Approximately 10 mg of chlorfluazuron standard was accurately weighed and dissolved in methanol to obtain a 1000 mg/L standard solution, which was kept at 4 °C in the dark. Appropriate amounts of standard chlorfluazuron solution were accurately transferred and diluted with the blank vegetable matrix extract to prepare the matrix-matched standard solutions of 0.002, 0.01, 0.05, 0.1, 0.5, and 1.0 mg/L for sample quantification.

### 2.2. Instruments

The samples were analyzed using ACQUITY UPLC Xevo TQ-S micro (Waters, Milford, MA, USA); Waters CORTECS UPLC C18 column (2.1 mm × 50 mm, particle size: 1.6 μm); column temperature: 40 °C; injection volume: Five μL; mobile phase: Phase A: Water (0.02% formic acid + 5 mmol/L ammonium acetate), phase B: Acetonitrile. The gradient elution procedure is shown in Table 1.

Ionization mode: ESI^+^; capillary voltage was 1.5 KV; ion source temperature was set at 150 °C; desolvation gas and cone gas were high-purity nitrogen, with flow rates of 600 L/h and 50 L/h, respectively. Collision gas was high-purity argon, with a flow rate of 0.18 L/min. Other mass spectrometry parameters are shown in Table 2.

### 2.3. Field Trial Tests

Field trial tests were performed in six different provinces in 2016 located in Anhui and Hubei provinces. The climate is humid and rainy. Each test plot was set as 30 m^2^, and per test was performed in triplicate. A blank control treatment was performed same as test treatments, and a protection line was built between each plot. The water spray method was used for pesticide application. The treatments are shown in Table 3. There were 2 kg sample collected in each treatment. All test sample residues were cryopreserved at −20 °C until use.

### 2.4. Sample Preparation

Five grams of each sample was added to 10 mL of water. The mixture was then vortexed with 20 mL of acetonitrile for 1 min, followed by ultrasonic extraction for 10 min. Then, 4 g of sodium chloride was added, and the mixture was centrifuged at 4500 rpm for 5 min. The upper layer of the acetonitrile extract was retrieved and further concentrated to dryness at 40 °C, and dissolved in 2 mL of acetonitrile. The acetonitrile solution was transferred to a solid phase extraction cartridge (NH_2_, 300 mg/3 mL) with 25 mL of acetonitrile/toluene (3 + 1, V/V) as eluent. The eluent was collected and concentrated to near dryness at 40 °C. The dried product was then dissolved in 2 mL of acetonitrile. The solution was filtered with a 0.22-μm filter for further testing.

### 2.5. Risk Assessment of Chronic Dietary Intake

The risk assessment of chronic dietary intake was calculated according to Equations (1) and (2):(1)NEDI=(∑STMRi×Fi)/bw
(2)RQ=NEDI/ADI

In Equation (1), NEDI is the estimated daily intake per person in China, μg/(kg bw·d); STMRi is the median value of the standard test residue of the i grade agricultural products, mg/kg, Fi is the population consuming the i grade dietary agricultural product, g.

In Equation (2), RQ (risk quotient) is the risk quotient, and ADI is the tolerable intake of pesticide per kilogram body weight, mg/kg bw. If RQ ≤ 1, then the risk is acceptable. A smaller RQ value suggests lower risk. If RQ > 1, then there is an unacceptable chronic risk. The larger RQ value implies greater risk.

## 3. Results and Discussion

### 3.1. Optimization of Analytical Method

The target ion scan was used to determine the presence of [M-H]^+^, and the detection of the target compound was optimized using the multiple reaction monitoring (MRM) mode. The chromatogram of chlorfluazuron is shown in Figure 1.

The pretreatment method for purifying was to use the solid phase adsorbent to adsorb the compound from the liquid phase, then to separate the target compound and the impurities in the sample, thereby achieving the separation and enrichment of the sample. This paper compared the purification effects of five different filling materials of the SPE cartridge (including C_18_, Florisil, NH_2_, carbon, Alumina-N) with different vegetable matrices. The optimal filling material was selected by comparing the recovery and the purification effect. The average recovery and matrix effects of the target compounds in different treatment groups are shown in Figure 2. When C_18_ and NH_2_ filling materials were used in the SPE cartridges, the recovery of chlorfluazuron was higher (93.4–87.6%), while the adsorption capacity of C_18_ was weak, and the color of the extract was darkest (Figure 3). The carbon filling material was the most appropriate for pigment adsorption, but the recovery of both carbon and Florisil was low (60.1% to 75.7%). Alumina-N had the lowest recovery for chlorfluazuron (46.8% to 55.7%). On the other hand, when we used the NH_2_ column, the matrix effect of the target compound was lowest in all cartridges. Therefore, the NH_2_ filling material was used in the SPE cartridge for vegetable sample purification.

### 3.2. The Linear Relationship and Limit of Quantification

The linear relationship and the limit of quantification (LOQ) of chlorfluazuron in the four vegetable matrices are shown in Table 4. Chlorfluazuron had a good linear relationship within the range of 0.01–10 mg/L, r^2^ = 0.9991–0.9997. An external standard was used for quantification. The LOQ of chlorfluazuron in four vegetable matrices was 0.01 mg/kg.

### 3.3. Accuracy and Precision

The test results of chlorfluazuron are shown in Table 1. The average recovery of chlorfluazuron in Chinese cabbage, Chinese kale, Chinese lettuce, and cauliflower was 89.2–104.1% ± 2.5–7.2% SD, 78.7–90.6% ± 2.3–9.6%, 76.5–93.2% ± 3.3–4.4%, and 75.0–85.5% ± 3.0–8.8% SD, respectively. The accuracy and precision of the method satisfied the requirements for pesticide residue analysis.

### 3.4. Dissipation Kinetics of Chlorfluazuron in Vegetables

The residue of chlorfluazuron in the edible parts of vegetables gradually decreased with time, and the dissipation process was of first-order reaction kinetics (Figure 4). The original deposition of chlorfluazuron in Chinese cabbage, Chinese kale, Chinese lettuce, and cauliflower was within the range of 0.625–6.072 mg/kg. After 28 days of application of pesticides, the residual dissipation rate was >90%, and the half-life was 2.4–12.6 d. The dissipation behavior at different testing sites were generally the same. The dissipation rate at the Hubei site was slightly faster than that of the Anhui site, and the half-life was shorter, which could be attributable to differences in meteorological conditions, (e.g., temperature, humidity, wind speed, and rainfall) among testing sites. Compared with the degradation rate of chlorfluazuron in tea (7.0-8.2 d), it degrades faster in Chinese cabbage, Chinese lettuce, and cauliflower [10].

### 3.5. Final Residue Testing

The results of final residue testing are shown below. Chlorfluazuron on Chinese cabbage and Chinese kale was assessed using the standard residue testing. When the last pesticide application interval was three days, and the residual amount of chlorfluazuron on Chinese cabbage and Chinese kale was 0.12–10.7 mg/kg (median: 1.08 mg/kg) and 0.19–8.06 mg/kg (median: 1.08 mg/kg), respectively. At five-day intervals, the residual amount of chlorfluazuron on Chinese cabbage and Chinese kale was <0.01–3.95 mg/kg (median: 0.62 mg/kg) and 0.06–5.48 mg/kg (median: 1.28 mg/kg), respectively. At seven-day intervals, the residual amount of chlorfluazuron on Chinese cabbage and Chinese kale was <0.01–2.83 mg/kg (median: 0.40 mg/kg) and <0.01–3.20 mg/kg (median: 0.51 mg/kg), respectively.

Chlorfluazuron on Chinese lettuce and cauliflower was assessed by the standard residue testing. When the last application interval was five days, the residual amount of chlorfluazuron on Chinese lettuce and cauliflower was <0.01–0.77 mg/kg (median: 0.18 mg/kg) and <0.01–1.54 mg/kg (median: 0.10 mg/kg), respectively. At seven-day intervals, the residual amount of chlorfluazuron on Chinese lettuce and cauliflower was <0.01–0.51 mg/kg (median: 0.10 mg/kg) and <0.01–1.06 mg/kg (median: 0.06 mg/kg). At the 14-day interval, the residual amount of chlorfluazuron on Chinese lettuce and cauliflower was <0.01–0.23 mg/kg (median: 0.06 mg/kg) and <0.01–0.92 mg/kg (median: <0.01 mg/kg), respectively.

### 3.6. Risk Assessment of Chronic Dietary Intake

The standard pesticide residue test was performed on four vegetables to detect chlorfluazuron, and the median residual values of chlorfluazuron in four vegetables were obtained. The value in Chinese cabbage was 0.40 mg/kg (seven days) and in Chinese kale was 0.51 mg/kg (seven days); the value in Chinese lettuce was 0.06 mg/kg (14 days) and in cauliflower was <0.02 mg/kg (14 days). According to the principle of maximum risk, 0.51 mg/kg was used as the median residual value of dietary risk. The ADI value of chlorfluazuron was 0.005 mg/kg bw. Based on the dietary components of different Chinese populations, NEDI and RQ were estimated. Table 5 indicates that the NEDI value of chlorfluazuron in different age categories and genders in Chinese populations was between 0.753–1.661 μg/(kg bw·d), and the RQ was 0.15–0.35. The NEDI and RQ were both < 1, indicating that the dietary intake risk of chlorfluazuron was low and falls within an acceptable range. And in India, harvesting cabbage at least on the 7th day after twice applications of chlorfluazuron (in the recommended dose) in all the locations could be safe for human consumption (adults) [11].

## 4. Conclusions

An UPLC-MS/MS method for the determination of chlorfluazuron in Chinese cabbage, Chinese kale, Chinese lettuce, and cauliflower was established using the optimized solid-phase extraction pretreatment technique in conjunction with the UPLC-MS/MS method. The analytical method demonstrated the advantages of convenient operation and high sensitivity. Based on the results of the standard residue test, the dissipation rate of chlorfluazuron in the edible parts of vegetables was rapid, indicating that chlorfluazuron is a readily degradable pesticide. Further assessment of the median values of the residues obtained in the final residue test indicated that the chlorfluazuron residue in vegetables consumed by the general Chinese population poses a relatively low health risk to humans.

## Figures and Tables

**Figure 1 ijerph-16-01758-f001:**
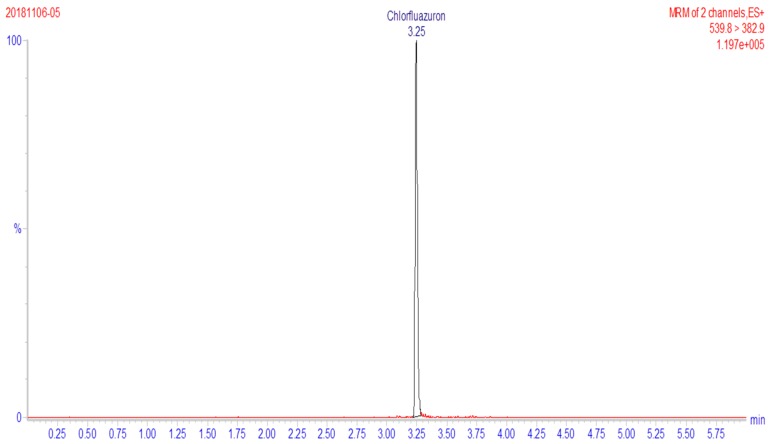
Chromatogram of chlorfluazuron.

**Figure 2 ijerph-16-01758-f002:**
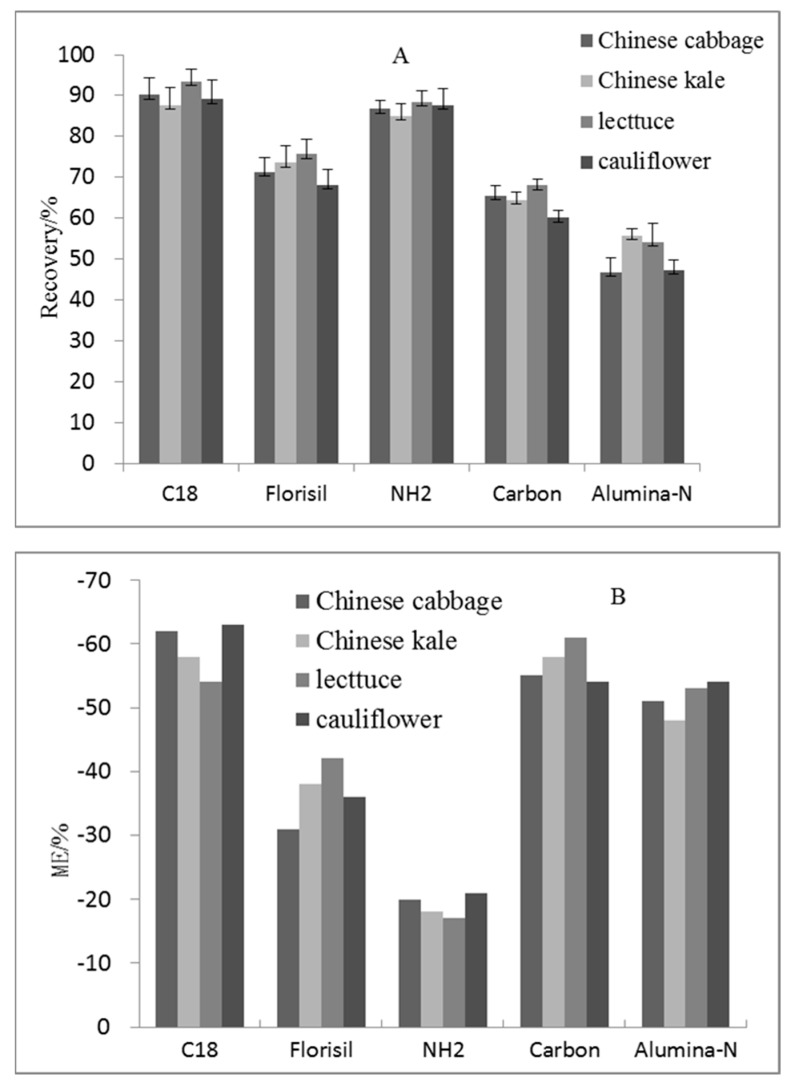
Effects of different SPE purification columns on chlorfluazuron recovery (A) and matrix effects (B) (*n* = 5).

**Figure 3 ijerph-16-01758-f003:**
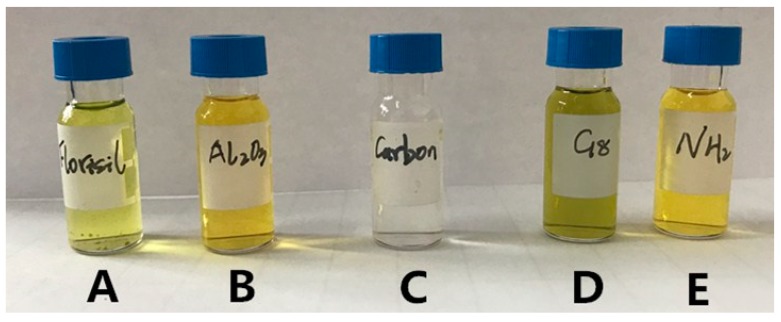
Colors of bok choy after treatment with different SPE purification columns (A: Florisil, B: Alumina-N, C: Carbon, D: C_18_, E: NH_2_).

**Figure 4 ijerph-16-01758-f004:**
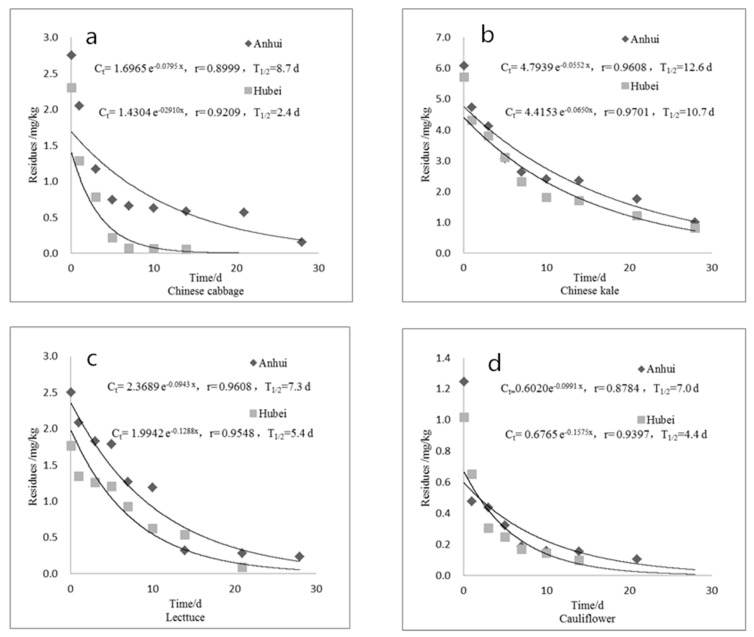
Dissipation kinetics of chlorfluazuron in Chinese cabbage (**a**); Chinese kale (**b**); lettuce (**c**); and cauliflower (**d**).

**Table 1 ijerph-16-01758-t001:** Gradient elution procedure.

Time (min)	Phase A: Water (0.02% Formic Acid + 5 mmol/L Ammonium Acetate)	Phase B: Acetonitrile
0.0	95%	5%
0.8	95%	5%
3.0	5%	95%
6.0	95%	5%

**Table 2 ijerph-16-01758-t002:** Mass spectrometry conditions for chlorfluazuron analysis.

Compound	Retention Time	Parent Ion	Daughter Ion	Cone Voltage	Collision Voltage
Chlorfluazuron	3.24	539.8	382.9 *	33	20
			158.0	33	20

* Indicated as the quantitative ion.

**Table 3 ijerph-16-01758-t003:** Experimental design of field trials.

Final Residue				Dissipation Kinetics (One-Time Pesticide Application)		
Site	Pesticide Dose (g a.i/ha)	Times	Sampling Intervals Since last Pesticide Use (d)	Site	Pesticide Dose (g a.i/ha)	Sampling Intervals (d)
Anhui, Beijing, Jilin, Chongqing, Hubei, Guangdong	45, 60	2, 3	3, 5, 7	Anhui, Hubei	60	2 h, 1, 3, 5, 7, 10, 14, 21, 28

**Table 4 ijerph-16-01758-t004:** Linear range, linear equation, correlation coefficient, recovery, and relative standard deviation (RSD) of chlorfluazuron.

Matrix	Linear Range (ng/mL)	Linear Regression Equation	Correlation Coefficient	Spiked Level	Average Recovery	RSD *
Chinese cabbage	0.01–10	y = 743,076x + 4601	0.9997	0.01	89.2	7.2
				1	96.3	6.5
				10	104.1	2.5
Chinese kale	0.01–10	y = 770,638x + 7,772	0.9995	0.01	78.7	4.5
				1	76.3	9.6
				10	90.6	2.3
Chinese lettuce	0.01–10	y = 811,458x + 9,853	0.9991	0.01	76.5	3.3
				1	95.4	4.4
				10	93.2	3.7
Cauliflower	0.01–10	y = 748,796x + 5,099	0.9993	0.01	75.0	8.8
				1	85.5	2.8
				10	85.5	3.0

* RSD means that relative standard deviation.

**Table 5 ijerph-16-01758-t005:** Risk assessment of chronic dietary intake of chlorfluazuron.

Age (y)	Gender	Weight (kg)	Vegetable Intake (g/d) [12]	Estimated National Intake (μg/(kg bw·d))	Risk qu otient
2–3	Male	13.2	43.0	1.661	0.33
	Female	12.3	39.6	1.642	0.33
4–6	Male	16.8	56.4	1.712	0.34
	Female	16.2	56.2	1.769	0.35
7–10	Male	22.9	70.2	1.563	0.31
	Female	21.7	65.9	1.549	0.31
11–13	Male	34.1	77.2	1.155	0.23
	Female	34.0	73.1	1.097	0.22
14–17	Male	46.7	87.1	0.951	0.19
	Female	45.2	81.5	0.920	0.18
18–29	Male	58.4	92.1	0.804	0.16
	Female	52.1	84.5	0.827	0.17
30–44	Male	64.9	93.7	0.736	0.15
	Female	55.7	91.3	0.836	0.17
45–59	Male	63.1	99.5	0.804	0.16
	Female	57.0	94.7	0.847	0.17
60–69	Male	61.5	97.7	0.810	0.16
	Female	54.3	93.2	0.875	0.18
≥70	Male	58.5	88.6	0.772	0.15
	Female	51.0	75.3	0.753	0.15

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
