# Peer review of "Determination of the Residue Behavior and Risk Assessment of Chlorfluazuron in Chinese Cabbage, Kale, Lettuce and Cauliflower by UPLC-MS/MS"

_ijerph, 2019, doi:10.3390/ijerph16101758_

Round 1
Reviewer 1 Report
This paper show the results of the determination of an harmful pesticide for human health (chlorfluazuron) in different vegetables by means of a procedure based on SPE extraction followed by UPLC-MS/MS analysis. Different type of SPE cartridges have been tested in order to choose the best one giving the highest recovery and purity.
The topic is of current relevant interest for a broad range of readers.
I suggest only few minor revisions:
- l. 96: "and THE DETECTION OF the target compound"
- l. 105: please provide more information for non-expert readers on the cartridges used. Which type of material is Florisil? Maybe this information can be expanded at l. 53.
- Figure 2: the quality of this figure must be improved
Author Response
Dear Editors and Reviewer:
Thank you for your letter and for the reviewer’s comments concerning our manuscript entitled “Determination of the residue behavior of chlorfluazuron in vegetables by UPLC-MS/MS” (ID: ijerph-491694). Those comments are all valuable and very helpful for revising and improving our paper, as well as the important guiding significance to our researches. We have studied comments carefully and have made correction accordingly, which we hope meet with approval. The main corrections in the paper and the responds to the reviewer’s comments are listed below point by point:
1) l. 96: "and THE DETECTION OF the target compound".
Answers:Thank you for your useful suggestion. We have revised this sentence.
2) l. 105: please provide more information for non-expert readers on the cartridges used. Which type of material is Florisil? Maybe this information can be expanded at l. 53.
Answers:Thank you for your suggestion. We have added the information of Florisil in the manuscript.
3) Figure 2: the quality of this figure must be improved
Answers:Thank you for your suggestion. We have revised Fig 2 in the manuscript.
I hope these revisions will make you accept my article. You can make any revision if you think it is appropriate. Thank you for your help. And I am looking forward to hearing from you.
Sincerely,
Jinsheng Duan

Reviewer 2 Report
Dear Authors;
1. abstract, line 25 .....and the risk quotient was 0.15 - 0.15 .....I think this is mistake;
2. materials and methods section line 58 - standard solution concentration is 1 mg/L or 1000 ug/L in the text is 1,000 mg/L;
3. Sample procedure section:authors should change this section because the description of sample preparation and purification using the SPE technique is not enough clearly
What does mean "near dryness" (lines 81 and 84)????
4. Results and discussion section: Discussion is very poor. Authors should compare their results with the results of other authors, because now authors only presented their results.
Author Response
Dear Editors and Reviewer:
Thank you for your letter and for the reviewer’s comments concerning our manuscript entitled “Determination of the residue behavior of chlorfluazuron in vegetables by UPLC-MS/MS” (ID: ijerph-491694). Those comments are all valuable and very helpful for revising and improving our paper, as well as the important guiding significance to our researches. We have studied comments carefully and have made correction accordingly, which we hope meet with approval. The main corrections in the paper and the responds to the reviewer’s comments are listed below point by point:
1) 1. abstract, line 25 .....and the risk quotient was 0.15 - 0.15 .....I think this is mistake;.
Answers:Thank you for your useful suggestion. We have revised the value in the manuscript.
2) materials and methods section line 58 - standard solution concentration is 1 mg/L or 1000 ug/L in the text is 1,000 mg/L
Answers:Thank you for your suggestion. The concentration of standard store solution is 1000 mg/L, and the highest concentration of standard work solution is 1 mg/L.
3) Sample procedure section:authors should change this section because the description of sample preparation and purification using the SPE technique is not enough clearly What does mean "near dryness" (lines 81 and 84)????.
Answers:Thank you for your suggestion. We have revised this section to state clearly.
4) Results and discussion section: Discussion is very poor. Authors should compare their results with the results of other authors, because now authors only presented their results.
Answers:Thank you for your useful suggestion. We have added the discussion to compare the degradation rate in the manuscript.
I hope these revisions will make you accept my article. You can make any revision if you think it is appropriate. Thank you for your help. And I am looking forward to hearing from you.
Sincerely,
Jinsheng Duan

Reviewer 3 Report
The manuscript presents a study of the residual behavior of chlorfluazuron in 4 types of vegetables and a risk assessment of its dietary intake. The results are interesting for toxicity studies. However, some serious flaws require attention before publication, for example:
- In title, replace "vegetables" by "Chinese cabbage, kale, lettuce and cauliflower".
- In abstract, remove first sentence, since the "low-toxicity" of CLFZ. Reconsider the second one.
- Some relevant references are missing. For instance: Chen et al. (Food Control, 2012, doi: 10.1016/j.foodcont.2011.11.027) and Ganguly et al. (Environmental Toxicology and Chemistry, 2017, doi:10.1002/etc.3872). Chen et al presented results on 2 pyrethroid pesticides and 3 benzoylurea pesticides in tea. Ganguly et al presented results on the persistence of CLFZ in cabbage and its risk assessment.
- In accordance with the above, discussion must be extended, including residual behavior, analytical method and results of risk assessment.
- please include exact size of test and control plots.
- information contained in supplementary tables cannot be supplementary. Add to the text.
- Information on field experiments must be extended, seems scarce (climatic conditions, sample size...)
- Where does the vegetable intake comes from? any reference? It is then assumed that all the vegetables consumed belong to this 4 types?
- In general, discussion must be extended, since there is absolutely no reference to the existing literature in Results and Discussion.
- Should not the term "risk assessment" be present in the title?
Author Response
Dear Editors and Reviewer:
Thank you for your letter and for the reviewer’s comments concerning our manuscript entitled “Determination of the residue behavior of chlorfluazuron in vegetables by UPLC-MS/MS” (ID: ijerph-491694). Those comments are all valuable and very helpful for revising and improving our paper, as well as the important guiding significance to our researches. We have studied comments carefully and have made correction accordingly, which we hope meet with approval. The main corrections in the paper and the responds to the reviewer’s comments are listed below point by point:
1) In title, replace "vegetables" by "Chinese cabbage, kale, lettuce and cauliflower"..
Answers:Thank you for your useful suggestion. We have revised the title according to your suggestion..
2) In abstract, remove first sentence, since the "low-toxicity" of CLFZ. Reconsider the second one.
Answers:Thank you for your suggestion. We have removed the “low-toxicity” in first sentence.
3) Some relevant references are missing. For instance: Chen et al. (Food Control, 2012, doi: 10.1016/j.foodcont.2011.11.027) and Ganguly et al. (Environmental Toxicology and Chemistry, 2017, doi:10.1002/etc.3872). Chen et al presented results on 2 pyrethroid pesticides and 3 benzoylurea pesticides in tea. Ganguly et al presented results on the persistence of CLFZ in cabbage and its risk assessment.
Answers:Thank you for your suggestion. We have added the references according to your suggestion.
4) In accordance with the above, discussion must be extended, including residual behavior, analytical method and results of risk assessment.
Answers:Thank you for your useful suggestion. We have revised it according to your suggestion.
5) please include exact size of test and control plots.
Answers:Thank you for your useful suggestion. We have added the exact size of test and control plots in the manuscript.
6) information contained in supplementary tables cannot be supplementary. Add to the text.
Answers:Thank you for your useful suggestion. We have revised the title of table in the manuscript.
7) Information on field experiments must be extended, seems scarce (climatic conditions, sample size...)
Answers:Thank you for your useful suggestion. We have added the information of field experiments in the manuscript.
8) Where does the vegetable intake comes from? any reference? It is then assumed that all the vegetables consumed belong to this 4 types?
Answers:Thank you for your useful question. The data of the vegetable intake is source from the document of “Chinese residents' dietary nutrient reference intake”. And the data was used for the 4 types.
9) In general, discussion must be extended, since there is absolutely no reference to the existing literature in Results and Discussion.
Answers:Thank you for your useful suggestion. We have added the discussion to compare the degradation rate in the manuscript.
10) Should not the term "risk assessment" be present in the title?
Answers:Thank you for your useful suggestion. We have added the “risk assessment” in the title.
I hope these revisions will make you accept my article. You can make any revision if you think it is appropriate. Thank you for your help. And I am looking forward to hearing from you.
Sincerely,
Jinsheng Duan

Reviewer 4 Report
Interesting project relating to the evaluation of the residual dissipation pattern and risk for dietary intake of chlorfluazuron in various vegetables (Chinese cabbage, Chinese kale, Chinese lettuce, and Cauliflower), using a solid phase extraction-ultra performance liquid chromatography-tandem mass spectrometry method. However some issues arise from the analysis of the paper which are presented in more detail below.
Relatively general Introduction section; develop with more detail the framework of the work.
Review and improve the text all over the document. For example, the sentence:
Lines 101-103: “The pretreatment method for purifying the SPE cartridge was to use the solid phase adsorbent to adsorb the compound from the liquid sample to realize the separation of the target compound and the impurities in the sample, thereby achieving sample separation and enrichment”
Improve the quality of the equations 1 and 2 (digitalized images?) and the resolution of the Figures.
Line 77: Sample Procedure? Possibly “Sample Preparation”.
Table 1 – the units are missing in the “Linear range”; remove “y=” from the header of the third column; R2 means R2? In this case it is not the correlation coefficient, but the determination coefficient.
How the recoveries above 100% are explained?
Explain better the results in Figure 2b (matrix effect) were obtained.
Figure 4 – Identify each plot with a), b), c) and d) and assign each vegetable source to the former identification in the legend. Where did these data come from? What does Hubei and Anhui mean?
What is the physical meaning of a first order decaying rate? What can you really conclude from that? How do you explain such different decaying results especially for Chinese cabbage and Cauliflower?
Explain this claim in the Abstract, how do “The results of this study provide supporting data and empirical basis for guiding the reasonable use of chlorfluazuron in vegetable production and in evaluating its dietary intake risk in vegetables”?
Author Response
Dear Editors and Reviewer:
Thank you for your letter and for the reviewer’s comments concerning our manuscript entitled “Determination of the residue behavior of chlorfluazuron in vegetables by UPLC-MS/MS” (ID: ijerph-491694). Those comments are all valuable and very helpful for revising and improving our paper, as well as the important guiding significance to our researches. We have studied comments carefully and have made correction accordingly, which we hope meet with approval. The main corrections in the paper and the responds to the reviewer’s comments are listed below point by point:
1) Relatively general Introduction section; develop with more detail the framework of the work.
Answers:Thank you for your useful suggestion. We have added the statement to describe the framework of the work.
2) Review and improve the text all over the document. For example, the sentence:
Lines 101-103: “The pretreatment method for purifying the SPE cartridge was to use the solid phase adsorbent to adsorb the compound from the liquid sample to realize the separation of the target compound and the impurities in the sample, thereby achieving sample separation and enrichment”
Answers:Thank you for your suggestion. We have revised this sentence to “The pretreatment method for purifying was to use the solid phase adsorbent to adsorb the compound from the liquid phase, then to separate the target compound and the impurities in the sample, thereby achieving the separation and enrichment of sample”.
3) Improve the quality of the equations 1 and 2 (digitalized images?) and the resolution of the Figures.
Answers:Thank you for your suggestion. We have revised the equations 1 and 2.
4) Line 77: Sample Procedure? Possibly “Sample Preparation”.
Answers:Thank you for your useful suggestion. We have revised it according to your suggestion.
5) Table 1 – the units are missing in the “Linear range”; remove “y=” from the header of the third column; R2 means R2? In this case it is not the correlation coefficient, but the determination coefficient.
Answers:Thank you for your useful suggestion. We have revised Table1 according to your suggestion.
6) How the recoveries above 100% are explained?
Answers:Thank you for your useful question. The reason of the higher recoveries perhaps that the matrix effects and the non-target compound in test solution.
7) Explain better the results in Figure 2b (matrix effect) were obtained.
Answers:Thank you for your useful question. We have explained the Figure 2b in section 3.1.
8) Figure 4 – Identify each plot with a), b), c) and d) and assign each vegetable source to the former identification in the legend. Where did these data come from? What does Hubei and Anhui mean?
Answers:Thank you for your useful suggestion. We have revised the Figure 4, and the Hubei and Anhui means that the location of sample collecting.
9) What is the physical meaning of a first order decaying rate? What can you really conclude from that? How do you explain such different decaying results especially for Chinese cabbage and Cauliflower?
Answers:Thank you for your useful question. The rate of degradation depends on the environment and the crop. The difference of decaying between Chinese cabbage and cauliflower perhaps that the crop growth period. We study the degradation rate of pesticides to provide data support for pesticide safety evaluation and rational application.
10) Explain this claim in the Abstract, how do “The results of this study provide supporting data and empirical basis for guiding the reasonable use of chlorfluazuron in vegetable production and in evaluating its dietary intake risk in vegetables”?
Answers:Thank you for your useful suggestion. We have added the explanation in the Abstract.
I hope these revisions will make you accept my article. You can make any revision if you think it is appropriate. Thank you for your help. And I am looking forward to hearing from you.
Sincerely,
Jinsheng Duan

Round 2
Reviewer 2 Report
I have not any comments. I agree to publish this work in presence form.
Author Response
Dear Editors and Reviewer:
Thank you for your letter and for the reviewer’s comments concerning our manuscript entitled “Determination of the residue behavior of chlorfluazuron in vegetables by UPLC-MS/MS” (ID: ijerph-491694). Those comments were all valuable and very helpful for revising and improving our paper, as well as the important guiding significance to our researches. We are very pleased that you can accept our revision.
Sincerely,
Jinsheng Duan

Reviewer 3 Report
Authors have revised and corrected the manuscript, following almost 100% reviewer's recommendations. Still, the language should be revised carefully since some sentences make no sense and are poor in grammar (see for example sentences in: lines 26-28, 48-50, 82-84, ...and so on).
Also, some minor comments:
- "Supplementary Materials" section, should be deleted.
- authors responded that "data of the vegetable intake is source from the document of “Chinese residents' dietary nutrient reference intake”. And the data was used for the 4 types". Then, this reference should be included.
- In introduction: "However, the residual behavior, analytical method and results of risk assessment of chlorfluazuron in vegetables and risk assessment of dietary intake in humans have not been reported to date." This statement should be revised (poor English and grammar) and reconsidered. Actually, residual behaviour studies imply the existence of analytical method. Also, risk assessment in vegetables has been provided by other authors, see for instance reference [11].
- And more importantly, discussion must be improved. There is absolutely no reference to the existing literature in Results and Discussion.
Author Response
Dear Editors and Reviewer:
Thank you for your letter and for the reviewer’s comments concerning our manuscript entitled “Determination of the residue behavior of chlorfluazuron in vegetables by UPLC-MS/MS” (ID: ijerph-491694). Those comments are all valuable and very helpful for revising and improving our paper, as well as the important guiding significance to our researches. We have studied comments carefully and have made correction accordingly, which we hope meet with approval. The main corrections in the paper and the responds to the reviewer’s comments are listed below point by point:
1) Authors have revised and corrected the manuscript, following almost 100% reviewer's recommendations. Still, the language should be revised carefully since some sentences make no sense and are poor in grammar (see for example sentences in: lines 26-28, 48-50, 82-84, ...and so on).
Answers:Thank you for your useful suggestion. We have revised these sentences in the manuscript.
2) "Supplementary Materials" section, should be deleted.
Answers:Thank you for your suggestion. We have removed the “Supplementary Materials”.
3) authors responded that "data of the vegetable intake is source from the document of “Chinese residents' dietary nutrient reference intake”. And the data was used for the 4 types". Then, this reference should be included.
Answers:Thank you for your suggestion. We have added the references according to your suggestion.
4) In introduction: "However, the residual behavior, analytical method and results of risk assessment of chlorfluazuron in vegetables and risk assessment of dietary intake in humans have not been reported to date." This statement should be revised (poor English and grammar) and reconsidered. Actually, residual behaviour studies imply the existence of analytical method. Also, risk assessment in vegetables has been provided by other authors, see for instance reference [11].
Answers:Thank you for your useful suggestion. We have revised the description here seriously.
5) And more importantly, discussion must be improved. There is absolutely no reference to the existing literature in Results and Discussion.
Answers:Thank you for your useful suggestion. We have added the discussion and references in the manuscript.
I hope these revisions will make you accept my article. You can make any revision if you think it is appropriate. Thank you for your help. And I am looking forward to hearing from you.
Sincerely,
Jinsheng Duan

Reviewer 4 Report
The authors address satisfactorily in the revised manuscript all the recommendations received. I appreciate the effort and invite the authors to check their work carefully before resubmission, particularly, taking consideration to the English proofreading that needs to be carry out. In my opinion, after this step, the manuscript will be ready for acceptance.
Author Response

(The authors gave the same response as above.)
